# Analysis of Changes in the Tensile Bond Strenght of Soft Relining Material with Acrylic Denture Material

**DOI:** 10.3390/ma14226868

**Published:** 2021-11-14

**Authors:** Magdalena Wyszyńska, Ewa Białożyt-Bujak, Grzegorz Chladek, Aleksandra Czelakowska, Rafał Rój, Agata Białożyt, Olaf Gruca, Monika Nitsze-Wierzba, Jacek Kasperski, Małgorzata Skucha-Nowak

**Affiliations:** 1Unit of Dental Material Sciences, Department/Institute of Prosthetic Dentistry and Dental Material Sciences, Medical University of Silesia in Katowice, 15 Poniatowskiego Street, 40-055 Katowice, Poland; abialozyt@sum.edu.pl; 2Department of Engineering Materials and Biomaterials, Faculty of Mechanical Engineering, Silesian University of Technology, Konarskiego 18A Street, 44-100 Gliwice, Poland; Grzegorz.Chladek@polsl.pl; 3Department/Institute of Prosthetic Dentistry and Dental Material Sciences, Medical University of Silesia in Katowice, 15 Poniatowskiego Street, 40-055 Katowice, Poland; aczelakowska@op.pl (A.C.); rafstoma@gmail.com (R.R.); ogruca@sum.edu.pl (O.G.); mnitsze@sum.edu.pl (M.N.-W.); protstom@sum.edu.pl (J.K.); 4Unit of Dental Propedeutics, Department of Conservative Dentistry with Endodontics, Division of Medical Sciences in Zabrze, Medical University of Silesia in Katowice, 15 Poniatowskiego Street, 40-055 Katowice, Poland; mskucha-nowak@sum.edu.pl

**Keywords:** dentures, denture lining, long-term lining, soft lining materials, silicone-based lining, acrylic-based lining, gerodontology, oral health

## Abstract

Abrasions and pressure ulcers on the oral mucosa are most often caused by excessive pressure or incorrect fitting of the denture. The use of soft relining materials can eliminate pain sensations and improve patient comfort. The main functional feature of soft elastomeric materials is the ability to discharge loads from the tissues of the mucosa. (1) Background: The aim of the work was a comparative laboratory study of ten materials used for the soft lining of acrylic dentures. (2) Methods: There were materials based on acrylates (Vertex Soft, Villacryl Soft, Flexacryl Soft) and silicones (Sofreliner Tough Medium, Sofreliner Tough Medium, Ufi Gel SC, GC Reline Soft, Elite Soft Relining, Molloplast). Laboratory tests include the analysis of the tensile bond strength between the relining material and the acrylic plate of the prosthesis. The tests were conducted taking into account 90-day term aging in the distilled water environment based on the methodology presented in the European Standard ISO 10139-2. (3) Results: After three months of observation, the highest strength of the joint was characterized by Flexacryl Soft acrylic, for which the average value was 2.5 MPa. The lowest average value of 0.89 MPa was recorded for the GC Reline Soft silicone material. Over time, an increase in the value of the strength of the combination of acrylic materials and a decrease in these values in the case of silicone materials was observed. (4) Conclusion: Each of the tested silicone materials showed all three types of damage, from adhesive to mixed to cohesive. All acrylic-based materials showed an adhesive type of failure. Time did not affect the type of destruction.

## 1. Introduction

The use of acrylic full and partial removable dentures can cause problems in the mouth, such as dryness, burning or pain, which are most often caused by mechanical damage to the mucosa [1,2,3]. The soft relining materials that cover the mucosa surface of the prosthesis are intended to support the prosthetic treatment and restore the physiological function of the stomatognathic system [1]. Relining, thanks to its flexible properties, absorbs the pressure of the denture plate, thanks to which it relieves pain and facilitates adaptation to new dentures. The flexible layer also reduces the traumatizing effect of the prosthesis on the muco-bone substrate by evenly distributing the chewing forces and better fitting the prosthesis [1,2].

Currently, the most commonly used soft lining materials can be divided according to their chemical structure into acrylic and silicone [4]. Acrylate materials consist of a powder containing poly(methyl methacrylate) and the initiator of the polymerization process, which is benzoyl peroxide. The fluid contains methyl, ethyl and butyl methacrylate monomers with the addition of ethyl acetate [5]. The acrylic material undergoes a plasticization process that ensures the elastic properties of the material. There are externally and internally plasticized acrylic masses. In the first group, the most common plasticizer is dioctybutyl phthalate, which does not enter into a permanent bond with the base material, but only loosely connects with the polymer chain by means of a secondary bond. The main problem with the use of this group of acrylic materials is its hardening, directly caused by washing out of the plasticizer in the oral cavity environment. The material, showing worse flexibility, loses its therapeutic properties [6]. Moreover, phthalic acid esters released into the oral cavity environment may pose a potential health hazard. According to Matsumoto M. et al., these compounds may be harmful to human reproduction and development [7]. On the other hand, in the group of internally plasticized acrylics, the water-insoluble butyl ester of methacrylic acid is incorporated into the structure of the base material. The formation of the copolymer by means of primary bonds ensures the desired degree of elasticity of relining and durability of the material [1,5,6]. Acrylic resins can also be classified into hot-cured materials, which require indirect relining with the involvement of a dental technician and room temperature polymerized direct reloading materials. The materials polymerized at room temperature are in the form of a gel, which, after being placed on the denture plate, is shaped directly in the patient’s mouth. The first group includes, among others: Vertex Soft (Vertex Dental), Villacryl Soft (Zhermack), Flexacryl Soft (Lang Dental). In turn, the second group is dominated by materials for biological tissue regeneration, such as ViscoGel (De Trey), Ivoseal (Ivoclar), Kerr Fitt (Kerr Dental), and Coe Soft (GC) [8,9].

Silicone materials do not contain a plasticizer, and their elastic properties are determined by the internal structure. Silicones can polymerize under the influence of heat (HTV) or at room temperature—self-polymers (RTV). The group of hot polymerizing silicones has better mechanical properties and greater adhesion to acrylic material, including: Molloplast B (Detax), Mollomed (Dentax), and Flexor (Schutz Dental). Conversely, in the group of cross-linking masses at room temperature there are: Mollosil Plus (Detax), Ufi Gel SC (VOCO), GC Reline Soft (GC), SofrelinerTough (Tokuyama), and Elite Soft Relining (Zhermack). The basic component of silicones is polydimethylsiloxane with hydroxyl groups and a cross-linking agent. The catalyst is composed of dibutyl dilaurate in chemically hardened masses and benzoyl peroxide in thermoset plastics. Silicone materials show less water sorption, are characterized by a substantially stable hardness during use and physiological indifference. However, due to the lack of chemical affinity to the acrylic plate of the prosthesis, it is necessary to use special primers. The inferior quality of the connection with acrylic materials used to make denture plates may result in a gradual detachment of the relining from the denture after only a few months of use. This favors the deposition of food debris that is difficult to remove by the patient and the rapid deposition of bacterial plaque [5,6,10,11,12,13].

The use of lining materials is not a flawless method. However, it is often the only choice in certain clinical cases. Particularly noteworthy are clinical cases of patients with scleroderma, patients after surgical procedures in the head and neck area, as well as a large group of edentulous patients. The relining materials are intended to restore the proper function of the stomatognathic system, and the therapeutic effect is largely influenced by the patient’s attitude and commitment, as well as the specificity of the material. The satisfactory healing result of using flexible lining depends on the proper oral hygiene of patients. It is important that check-ups take place regularly and that the patient is under constant dental care. The dentist should carry out hygiene instructions already at the first visit and set an important goal, which is keeping the oral cavity clean and preventing the deposition of denture plaques [14,15,16]. The denture should be cleaned at least twice a day with a soft brush and low-abrasion toothpaste or soap. It is recommended to use mouth fluids daily, clean the tongue, wash hands thoroughly before removing and inserting dentures, and massage the gums with a soft brush. After each meal, the prosthesis should be cleaned under a stream of warm water. Dentures should be removed overnight and stored in a dry box after washing. Additionally, once a week, disinfect for 30 min in a 0.2% chlorhexidine solution. The prosthesis should not be placed in boiling water, nor should any agents containing a solution of hypochlorite and peroxides be used, as preparations of this type may have an adverse effect on the elastic lining material [17,18]. If, at the control visit, a gap exceeding 5 mm is noticed between the denture plate and the relining material, the soft material should be removed and the relining procedure should be repeated [6,19].

Based on a review of the literature, it has been shown that the most common causes of a negative assessment of soft relines in clinical conditions are their colonization by microorganisms, changes in hardness and detachment of the material from the denture plate. The speed with which these processes take place depends on the different conditions of the oral cavity environment in individual patients, methods of cleaning and storing dentures, eating habits, and the use of stimulants [9,10,19,20,21,22,23]. Changes in the parameters of the lining materials occurring during their use in the oral cavity, such as the durability of their connection with the denture plate, the formation of cracks on the surface, changes in hardness, also have a huge impact.

The assumption of the work was to identify materials for relining with the best properties in terms of the strength of the bond with acrylic material. The material that loses its connection with the denture plate during use loses its therapeutic properties. From the clinical point of view, one of the most important parameters for denture relining is the proper bonding of the relining material to the acrylic denture plate and its stability during use. This assessment determines the period in which relining performs its therapeutic role. The treatment should result in an improvement in functioning, comfort, and social well-being [24,25]. According to clinical trials, the use of soft relining materials in the study group significantly improved the quality of life of patients [26] and also significantly reduced the adaptation time to new dentures [27]. Similar results were obtained by Kimoto et al. [28], who compared conventional acrylic prostheses with a soft lining in terms of the condition of the prosthetic base and the presence of pain.

The undertaken research topic meets the unresolved problems of dental prosthetics and the social demand for proper functioning, and at the same time economical, solutions for prostheses. The relining materials were assessed in terms of the strong bond with the acrylic material, which is of fundamental importance due to the functional functions performed. The tested parameter directly determines the chewing efficiency and quality of life of the patients. The obtained results may contribute to the creation of an objective procedure for assessing the quality of the functioning of these materials based on laboratory and clinical criteria.

The aim of the study is to assess the tensile bond strength of soft relining materials with acrylic denture material and to determine the type of damage caused during the bond strength test, as well as to assess the impact of aging time on the type of destruction.

## 2. Materials and Methods

Ten materials used for long-term soft relining of removable dentures were selected for the research.

Tested silicone-based materials:Sofreliner Tough M (Tokuyama, Taitou-ku Tokyo, Japan) is a soft material based on addition silicone;Sofreliner Tough S (Tokuyama, Taitou-ku Tokyo, Japan) is an additional silicone-based material. The manufacturer classified the product as a super soft material;flexible type A silicone material that polymerizes at low temperature;Ufi Gel SC (VOCO, Cuxhaven, Germany) is a silicone A-based soft material;GC Reline Soft (GC, Tokyo, Japan) is a soft A-silicone-based relining material for long-term direct and indirect relining at room temperature;Elite Soft Relining (Zhermack, Badia Polesine, Italy) is an additive silicone material;Molloplast B (Detax, Ettlingen, Germany) is a one-component silicone used in the indirect method.

Acrylic-based materials tested:Vertex Soft (Vertex Dental, Soesterberg, The Netherlands) is a heat-polymerizing denture material;Villacryl Soft (Zhermack, Badia Polesine, Italy) is an acrylic material;Flexacryl Soft (Lang Dental, IL, USA) is a flexible acrylic material used in the indirect method for relining denture plater.

The test of the tensile bond strength of the lining materials with the acrylic material were made in accordance with the European standard ISO 10139-2 [29]. In the study, Vertex Rapid Simplified acrylic resin (Vertex-Dental B.V., The Netherlands) was used as the acrylic material used as a sample of the denture plate. This material is heat polymerizable and is used to make complete and partial dentures. Vertex Rapid Simplified was used to make plates with dimensions of 7 cm × 7 cm and a thickness of 3.5 mm, in accordance with the manufacturer’s recommendations. The plates were wet pre-ground on both sides with a standard metallographic grinder on 120 grit abrasive paper (Struers A/S, Copenhagen, Denmark) to remove unevenness. At this point, the thickness of the samples was 3.2 ± 0.2 mm. Particular attention was paid to grinding scratches, which should only run in one direction. Successively, the plates were rinsed, rotated by 90°, and the prepared surface was sanded with 220 and 320 grit abrasive paper successively until the scratches on the previous paper were no longer present. Then the thickness of the samples was checked, and it was (3.1 ± 0.2 mm).

Square samples with a side length of 25 ± 3 mm were cut from the plates, and then the roughness formed at the edges was ground. The samples were rinsed again, and this time, only the working surface was sanded on 500 grit sandpaper until the scratches from the previous sanding disappeared. The samples were rinsed again without touching the prepared surface, and the opposite surface was marked with a black marker to facilitate later identification. The acrylic samples prepared in this way were placed in distilled water at a temperature of 37 ± 1 °C for 28 days ± 5 h. After aging, the samples were removed from the bath in pairs, and the surface was dried from visible moisture. In the case of silicone materials, a binding agent dedicated to a given material was applied to the working surface of acrylic samples with an applicator. The first sample was placed on a compression table in a testing machine (Zwick/Roel, Germany). A polyethylene ring with an internal diameter of 11 mm and a height of 3 ± 0.2 mm was placed in the center of the sample, and the tested relining material was applied inside. Then a second acrylic plate was placed and pressed with a second punch with a force of 30 N, which was held until the material was cured.

In the case of acrylic materials, the procedure was different due to the method of polymerization of the material recommended by the manufacturer. It was necessary to use gypsum-filled polymerization can halves, between which the samples were placed. The whole was squeezed on a pneumatic press and transferred to the polymerization frames, after which the polymerization was carried out in the polymerizer.

Handles with a thickness of 3 ± 0.2 mm were attached to the polymerized samples in order to ensure the axial alignment of the samples in the jaws of the testing machine. The handles were attached in two stages. First, three samples were placed on the compression table mounted in the lower part of the working area of the machine, and three plates were mounted in the upper jaw of the machine. Vertex Castapress (Vertex-Dental B.V., The Netherlands) was used to install the tile holders. This material is a “self-curing” acrylic material intended for the production of full and partial dentures, repairs, relining, rebasing, and supplementing skeletal dentures. The Vertex Castapress resin was placed on the samples, and they were driven manually in such a way that the acrylic plates were immersed in the self-curing acrylic and, at the same time, did not touch the samples. The samples were then removed from the jaws, the compression table was removed, and the lower stretching jaws were mounted. Specimens were fixed in the lower jaws, and in the upper jaws, another three acrylic plates were holders. The procedure for polymerizing the handles was repeated. After the acrylic had hardened, the samples were transferred to a water bath at 37 ± 1 °C. Three aging times were used: 7 days ± 1 h, 28 days ± 2 h, and 90 days ± 2 h. Ten samples were made for each compilation tested material. After removing them from the bath, the samples were clamped in the jaws of a testing machine and stretched at a speed of 10 mm/min until the sample was completely ruptured (Figure 1).

For each of the samples, the type of damage that occurred during the experiment was visually determined [10]:*A*—adhesive or interfacial failure—consists in separating the lining material and acrylic;*C*—cohesive failure—consists in breaking the soft lining material;*M*—mixed failure, which is a compilation of adhesive and cohesive.

Then, the bond strength *σ_B_* (MPa) was calculated on the basis of the relationship:$$\sigma_{B}=\frac{F_{\max}}{A}$$
where:

*F*_max_—maximum recorded force expressed in N;

*A*—initial area of the silicone connection or the cross-section of silicone layers in a plane parallel to the base of the sample with acrylic, in practice the area determined by the internal diameter of the polyethylene ring, expressed in mm^2^.

Statistical analysis was conducted using the PQStat ver. 1.6.6.204 (PQStat Software, Poland). The data obtained during laboratory tests were used for calculations aimed at finding statistically significant relationships and comparing the results of individual materials used for long-term, soft acrylic dentures relining. The level of significance was α = 0.05. The test results were analyzed of variance (ANOVA) for one-factor systems (α = 0.05) with a possible F * correction (Brown–Forsythe) when the assumption of equal variance was not met. The tests were preceded by checking the assumption of homogeneity of variance with Levene’s test. In case of rejection of the null hypothesis, determined by the analysis of variance of the lack of equality between the means, the differences between the means of the individual groups were examined using the post hoc HSD Tukey test. This way, it was checked which of the n-means differ from each other and which are equal to each other. During the bond strength tests, the type of fractures was also determined. These results were also analyzed statistically. Due to the fact that the sample size was small, and thus the expected values were often less than five, the Fisher–Freeman–Halton exact test was used for the R × C tables (α = 0.05).

The obtained results were summarized and presented in the form of tables and figures.

## 3. Results

During the tests, the strength of the connection of the soft relining material with the material intended for denture plates was measured after 7 days, and the obtained values were compared with the requirements of ISO 10139-2. Compliance with the requirements of ISO 10139-2 [29] in terms of bond strength by individual materials in the context of their long-term use is shown in Table 1. The standard requirements were not met by Villacryl Soft, for which six samples had bond strength values lower than 1 MPa.

The results of the bond strength tests and statistical analysis are presented in Figure 2 and in Table 2.

After 7 days of conditioning in distilled water, a statistically significant influence of the lining material (*p* < 0.05) on the bond strength values was found (Table 2). The post hoc test showed statistically significant differences between the materials. Flexacryl Soft, for which the average value of *σ_B_* was 4.4 MPa, was characterized by a higher tensile bond strength than other materials. For Vertex Soft, Molloplast B, and GC Reline Soft materials, the mean *σ_B_* values did not differ statistically significantly and ranged from 2.2 MPa to 2.6 MPa. The mean values obtained for GC Reline Soft were the lowest and, at the same time, did not differ statistically significantly from the results obtained for Sofreliner Tough Medium, Elite Soft Relining, and Ufi Gel SC. Similarly, they did not differ statistically significantly in terms of the *σ_B_* Sofreliner Tough Medium, Mollosil Plus, Elite Soft Relining, and Ufi Gel SC values. The average values of *σ_B_* for these materials ranged from 1.4 MPa to 1.7 MPa. Only one of the materials classified as a super soft material—Sofreliner Tough Soft—was characterized by an average value of the bond strength of 1.04 MPa. This value did not differ in a statistically significant manner from the results obtained for Sofreliner Tough Medium (1.6 Mpa), Mollosil Plus (1.38 Mpa), and Villacryl Soft (0.98 Mpa), the latter having the lowest mean *σ_B_* of all tested relining materials.

After 28 days of conditioning in distilled water, a statistically significant influence of the relining material (*p* < 0.05) on the bond strength values was found (Table 2), while the values of the Fisher statistics indicate a smaller differentiation of the mean values. On the basis of the post hoc test, statistically significant differences between the materials were demonstrated. Flexacryl Soft, for which the average value of *σ_B_* was 3.4 Mpa, was characterized by a statistically significant higher bond strength than other materials. There were no statistically significant differences between the average values of the bond strength for Sofreliner Tough Medium, Mollosil Plus, Vertex Soft, Ufi Gel SC, GC Reline Soft, Elite Soft Relining, and Molloplast B, which ranged from 1.48 MPa to 2 MPa, with the highest values recorded for Vertex Soft and Ufi Gel SC. There were no statistically significant differences between the average values of the bond strength for Sofreliner Tough Medium, Sofreliner Tough Soft, Mollosil Plus, Villacryl Soft, GC Reline Soft, Elite Soft Relining, and Molloplast B, which ranged from 1.2 MPa to 1.73 MPa. After 7 days of conditioning, Sofreliner Tough Soft and Villacryl Soft were characterized by the lowest mean values of *σ_B_*.

After 90 days of conditioning in distilled water, is shown in Figure 2. A statistically significant influence of the relining material (*p* < 0.050 on the bond strength was found (Table 2). At the same time, the values of the Fisher statistics indicate a smaller differentiation of the mean values than in the case of the two shorter conditioning times. On the basis of the post hoc test, statistically significant differences between the tested materials were demonstrated. The highest bond strength was characterized by Flexacryl Soft, for which the average value of *σ_B_* was 2.5 MPa, and this value did not differ statistically significantly from the results obtained for Villacryl Soft (*σ_B_* = 2.18 MPa). There were no statistically significant differences between the average values of the tensile bond strength for Sofreliner Tough Medium, Mollosil Plus, Villacryl Soft, Vertex Soft, Ufi Gel SC, Elite Soft Relining, and Molloplast B, which ranged from 1.34 MPa to 1.84 MPa. The lowest mean value of *σ_B_*, 0.89 MPa, was recorded for GC Reline Soft, and this value did not differ statistically significantly from the results obtained for Sofreliner Tough Soft and Mollosil Plus.

The influence of the conditioning time on the tensile bond strength for the lining materials is presented in Table 2 and in Figure 2. For most silicone materials, no statistically significant influence of the conditioning time on the mean value of the bond strength was observed. In the case of two silicone materials, GC Reline Soft and Molloplast B, the average values of the bond strength decreased with the conditioning time of the samples, however, in the case of the first material, the bond strength values gradually decreased, and for the second material, these values stabilized after 30 days.

All the analyses of acrylic materials showed statistically significant changes in the value of the bond strength in relation to the initial values. For Villacryl Soft, these values increased with time, and for Vertex Soft and Flexacryl Soft, they decreased with time.

The percentage of failure obtained after the tensile bond strength test showed statistically significant differences for the type of material and statistically significant for particular materials after different storing times (Table 3 and Figure 3). All acrylate-based materials showed adhesive failure. For each of the silicone materials, different types of fractures were demonstrated, ranging from adhesive, through mixed, to cohesive. In many cases, for a specific silicone material and conditioning time, all three types of failure were recorded. Time did not statistically significantly affect the type of breakthrough for any of the analyzed silicone and acrylic materials.

## 4. Discussion

An indispensable criterion for the correct relining of the denture is to make a permanent connection of the soft relining material with the acrylic denture plate. Three methods are commonly used to test the strength of this joint: peel, shear, and tensile. The most popular and the most reliable method, thanks to the standardization of the European Standard ISO 10139-2 [29], is the tensile strength test. The test consists in stretching the sample axially at a constant speed at room temperature until it breaks completely. In accordance with the requirements of ISO 10139-2 [29], soft materials used for long-term relining of denture plates should have a bond strength higher than 1 MPa, while materials classified as super soft should have a value higher than 0.5 Mpa. The measurement is performed after 24 h of conditioning the sample in distilled water.

The results of our own research show that the highest bond strength, both after 7 and 90 days of observation, was obtained for the acrylic material Flexacryl Soft (average values 4.41 Mpa and 2.5 Mpa). The lowest average value after 7 days of observation was obtained for the acrylic material Villacryl Soft—0.98 Mpa. As the only one among all tested materials, it obtained an average value of less than 1 Mpa. However, in the course of the experiment, the strength of the Villacryl Soft material bond increased by 55%, amounting to 2.18 Mpa after 3 months. For the last of the tested materials based on Vertex Soft acrylic, the average values of the bond strength were 2.15 Mpa after 7 days and 1.51 Mpa after 90 days. A decrease in tensile bond strength may indicate problems related to water absorption, solubility, increased hardness of the acrylic material and internal stresses in the area of the junction of the relining and the denture plate. Research results available in the literature show that the particularly unfavorable effects of aging are stronger for acrylate-based materials than for silicone materials [30,31,32,33].

In our own research, the GC Reline Soft material stood out among all the tested silicone materials, which showed a high average value of the bond strength after a week of observation—2.15 Mpa. The research conducted by Kim et al. [34] shows that after 24 h, this material showed even higher values of 2.99 ± 0.43 Mpa. According to these results, the material met the requirements of ISO 10139-2, according to which the materials used for long-term soft relining of denture plates should have a bond strength higher than 1 Mpa. This value is tested 24 h after making the sample. However, the author’s own research after three months of observation shows that the strength of the connection of GC Reline Soft with acrylic material significantly decreased, finally reaching the value of 0.89 Mpa. The test results for the GC Reline Soft material, as well as the above-mentioned Villacryl Soft material, may suggest the need to extend the requirements of ISO 10139-2 with an additional test after 30 days of sample aging. This would allow a more objective assessment of long-term materials.

Among the silicone materials, the highest values were obtained for the Molloplast B material, the value of which was 2.45 Mpa after a week and 1.84 Mpa after 3 months, not significantly different from the Vertex Soft acrylic material. This is due to the different structures of hot-curing materials. When exposed to temperature, the elastomer bonding system chemically reacts with the polymethacrylate, which, according to many authors, is one of the most durable bonds among elastic materials [31,35,36,37]. Aydin et al. [31] investigated the tensile bond strength of three different soft relining materials. In his work, he compared acrylic-based material, room-temperature-curing material, and hot-curing material. The latter showed the highest value of bond strength throughout the three-month observation period. Additionally, Madan et al. [36] obtained significantly higher values of bond strength values of the Molloplast B vulcanized at high temperature compared to the Mollosil material polymerized at room temperature. The samples were divided into two groups. In the control group, the samples were stored for 24 h in distilled water at 37 °C. In turn, 2500 thermal cycles were performed in the research group, which consisted of soaking the samples alternately at 5 °C and 55 °C, in order to recreate the conditions in the oral cavity. The average tensile strength of the control and heat cycled samples are: Mollosil (6.82 kg/cm^2^ and 8.41 kg/cm^2^) and Molloplast-B (16.30 kg/cm^2^ and 13.67 kg/cm^2^). The Molloplast-B material showed a much higher tensile strength than the self-polymerizing Mollosil, regardless of the use of thermal cycling.

The analysis of the results obtained by the author shows that the remaining tested silicone materials did not show significant changes in the value of the tensile bond strength; however, in each case, a decrease in these values was observed with time. The decrease in the average values of the bond strength ranged from 2.9% (Mollosil Plus) to 14.6% (Elite Soft Relining) and was not as significant as in acrylic materials. Studies available in the literature show a decrease in the tensile bond strength between silicone material and acrylic denture with time [31,37]. This is due to the aging of the materials in the water environment and the related change in hardness, as well as water absorption and solubility over time. During the three-month observation, the strength of the connection between GC Reline Soft and acrylic material decreased by 59%. In the studies by Białożyt-Bujak et al. [23], the same material showed the highest increase in hardness of all the tested silicones within 3 months, amounting to 29%.

Another parameter tested was the type of fracture made after the bond strength test. The acrylate-based materials showed an adhesive type of failure, and in all the tests performed, the material was detached from the denture plate. However, during the examination of silicone materials, all types of damage occurred: adhesive, cohesive, and mixed. Time did not significantly affect the type of failure for any of the analyzed materials. The predominance of the adhesive type of failure occurred in the Ufi Gel SC material after 7, 30, and 90 days. Więckiewicz et al. [37] obtained similar results for this material. Mutluay et al. [32], for different denture base materials and Molloplast-B with dedicated bonding agents, registered different failure types (from 100% cohesive to 100% mixed). This shows that the connection created by the adhesive system is denture base materials-dependent. It should be noted that, when adhesive failures are noted, the connection created by the bonding agent can be considered as the weakest part of the connection. When dealing with mixed failure, connection strength and soft lining material are comparable, for according to cohesive failures we can tell only the strength of the material itself is, in fact, lower than the strength of the interface layer formed by the bonding; thus, in practice, the mixed and cohesive failures should be considered as especially promising [38].

All the results were obtained under laboratory conditions. The oral environment that affects the underlying material during the use of the prosthesis is different from optimal laboratory conditions. Compliance with the rules of hygiene, the method of storing dentures, the use of adhesives, eating habits, and the use of stimulants may have a significant impact on the tested material.

## 5. Conclusions

The study of the tensile bond strength of the relining materials with the acrylic denture base material showed that the bond strength values increased with time for Villacryl Soft, decreased for Vertex Soft and Flexacryl Soft, and were almost stable in the case of silicone materials, excluding GC Reline Soft and Mollopast B (decrease of *σ_B_*). The bond strength and its changes are therefore dependent on the specific material used. It cannot be assumed in advance how a material belonging to a material group will behave over time, although there is a good chance that the silicone will be much more stable in the simulated period. Each of the tested silicone materials showed all three types of failure, from adhesive, through mixed, to cohesive. All acrylic-based materials showed adhesive failure. Time did not affect the type of failure, excluding Sofreliner Tough Medium and Mollosil Plus.

## Figures and Tables

**Figure 1 materials-14-06868-f001:**
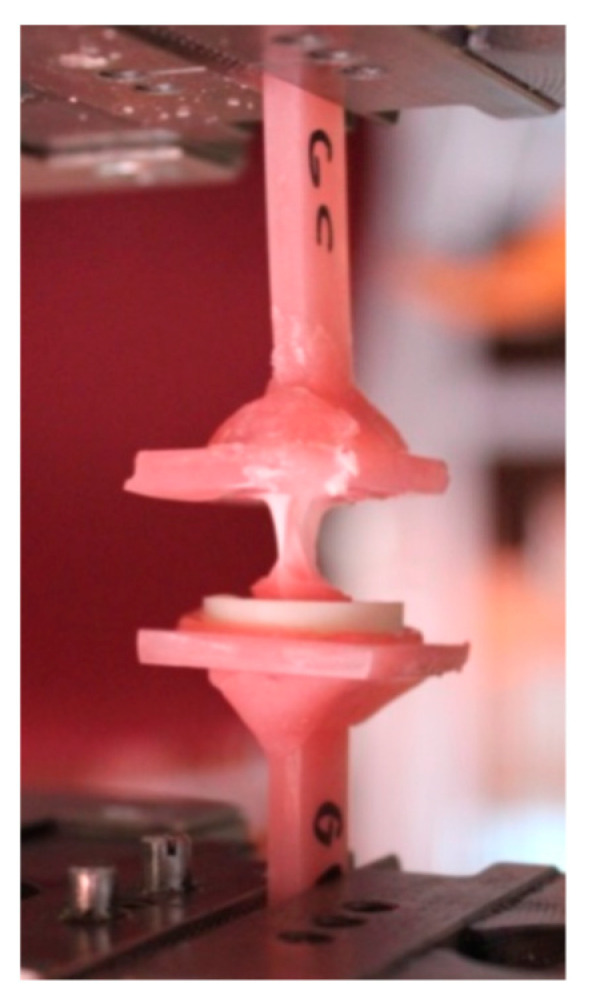
A sample during the test of the strength of the connection between the soft lining material and the acrylic resin.

**Figure 2 materials-14-06868-f002:**
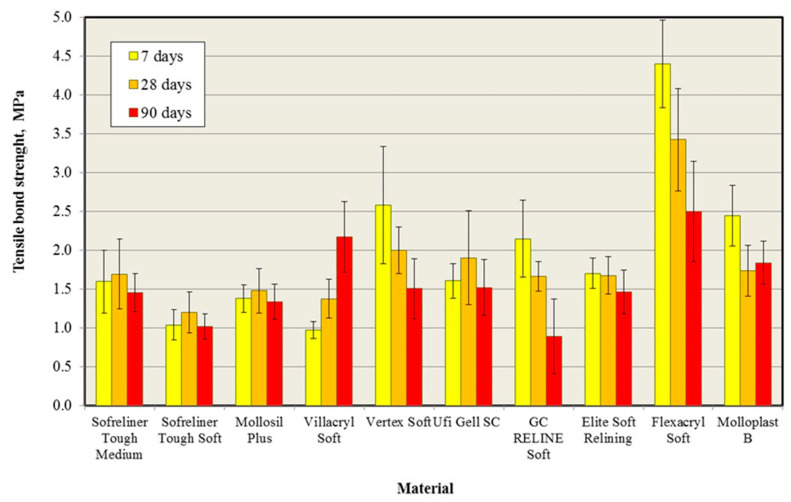
Average values of the tensile bond strength of the tested relining materials with acrylic material for denture plates during 90 days of storing in distilled water.

**Figure 3 materials-14-06868-f003:**
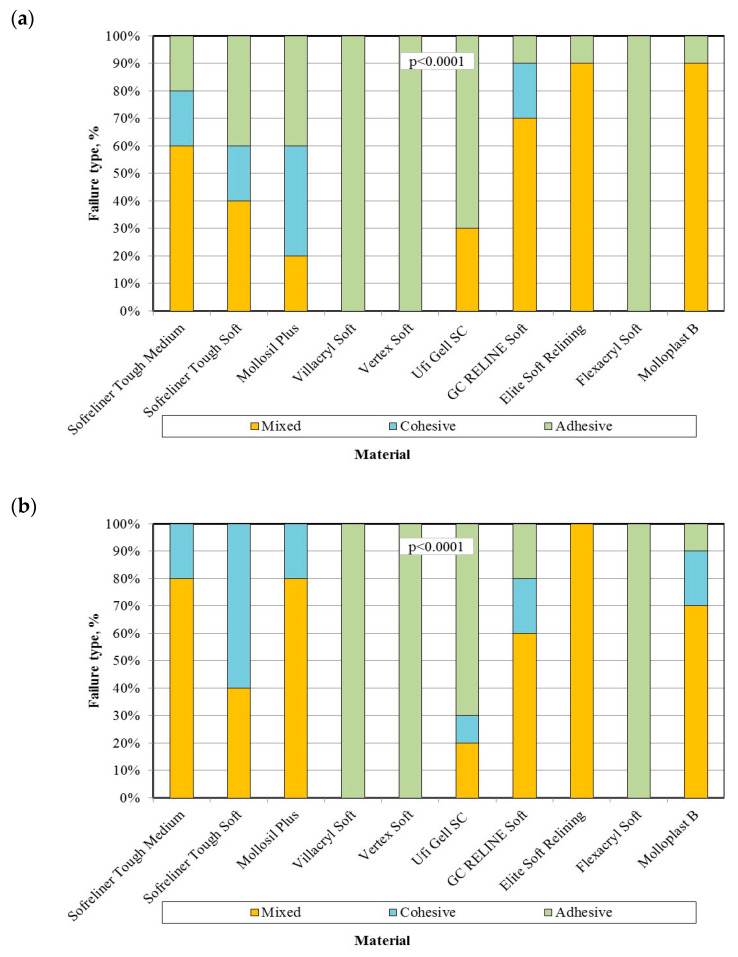
Influence of the lining material on the failure type in tensile strength test after 7 days (**a**), 28 days (**b**), and 90 days (**c**) of conditioning the samples in distilled water, α = 0.05.

**Table 1 materials-14-06868-t001:** Checking the degree of compliance with the requirements of ISO 10139-2 by the analyzed materials in terms of the strength of the connection with the material for denture plates.

Material	Number of Occurrences	
	Soft *σ_B_* < 1 MPa	Super Soft *σ_B_* < 0.5 MPa
Sofreliner Tough Medium	0	-
Sofreliner Tough Soft	-	0
Mollosil Plus	0	-
Villacryl Soft	6	-
Vertex Soft	0	-
Ufi Gel SC	0	-
GC Reline Soft	0	-
Elite Soft Relining	0	-
Flexacryl Soft	0	-
Molloplast B	0	-

*σ_B_*—bond strength for a soft liner material.

**Table 2 materials-14-06868-t002:** The results of one-way ANOVA and Tukey HSD post hoc tests for tensile bond strength *.

Storing Time/Days	Material									
	Sofreliner tough Medium (*p* = 0.405)	Sofreliner tough Soft (*p* = 0.284)	Mollosil Plus (*p* = 0.44)	Villacryl Soft (*p* ˂ 0.0001)	Vertex Soft (*p* = 0.0006)	Ufi Gell SC (*p* = 0.1593)	GC RELINE Soft (*p* ˂ 0.0001)	Elite Soft Relining (*p* = 0.0824)	Flexacryl Soft (*p* ˂ 0.0001)	Molloplast B (*p* = 0.0002)
7 (*p* = 0.0076)	a,b,d	a,c	a,b,c	A; c	A; e	a,b,d	A; d,e	b,d	A; f	A; e
28 (*p* = 0.0066)	a,b	A	a,b	B; a	A; b	b	A; a,b	a,b	B; c	B; a,b
90 (*p* = 0.0062)	a,c,d	a,c	a,c,d	C; b,d	B; a,d	a,d	B; e	a,c,d	C; b	B; d

* The different uppercase letters (A–C) for each column and lowercase letters (a–f) for each row show significantly different results at the *p* < 0.05 level.

**Table 3 materials-14-06868-t003:** The results of statistical analyses of failure types in tensile strength test changes for tested materials after 90-days experiment, α = 0.05.

Material	*p*-Value
Sofreliner Tough Medium	*p* = 0.0263
Sofreliner Tough Soft	*p* = 0.1981
Mollosil Plus	*p* = 0.0495
Villacryl Soft	*p* = 1
Vertex Soft	*p* = 0.1
Ufi Gell SC	*p* = 0.8566
GC RELINE Soft	*p* = 0.1203
Elite Soft Relining	*p* = 0.7537
Flexacryl Soft	*p* = 1
Molloplast B	*p* = 0.659

## Data Availability

The data presented in this study are available from the corresponding authors.

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
