# Peer review of "Analysis of Changes in the Tensile Bond Strenght of Soft Relining Material with Acrylic Denture Material"

_materials, 2021, doi:10.3390/ma14226868_

Round 1
Reviewer 1 Report
This study investigates the tensile strength of the connection of soft relining materials with resin prostheses.
The authors compare the characteristics of 10 materials, classified into two major families (acrylates and silicones) according to their chemical composition. They measured the tensile strength after 7, 28 and 90 days. They also compare the failure modes.
Introduction
The presentation of the different soft relining materials is particularly well done.
The major problematic point, which I invite the authors to revise, is that there is no consensus on the use of long-term flexible resin. While it is true that their use provides interesting comfort to patients, many clinicians are opposed to their permanent placement because it can lead to iatrogenic bone resorption. However, this does not detract from the interest of this work, as these materials are unanimously indicated for tissue conditioning or post-surgical healing. However, the introduction should be revised to make this distinction.
The absence of a long-term indication also modifies the reading of the results. If a practitioner uses soft materials in the short or medium term, it may be interesting to have a stable and retentive material with a hard resin base, within the limits of easy removal. Similarly, for the practitioner who wishes to remove the soft material, it is important to have an adhesive rather than a cohesive type of failure. Perhaps to be presented in the discussion.
Material and method
The methodology and images are very clearly presented.
Results
The presentation of the results is unnecessarily long. It is unuseful to detail the evolution of retention over time for each material. Similarly, a good legend for Graphs 3, 4, and 5 will reduce the length of the manuscript.
Can the authors suggest a more synthetic write-up of the results of their work?
Discussion
The discussion is interesting and of good quality in comparison with the existing literature.
Does the experimental set-up follow the recommendations of the ISO standard? Indeed, the construction of the specimens with multiple interfaces introduces many biases in the analysis of the results.
Why did the authors not use a thermocycling system to test the long-term behavior of the materials?
In conclusion, it would be interesting to put the results into perspective with the clinical use of soft lining materials (see introduction).
Reviewer 2 Report
The article showed a huge experimental effort and a lot of data that were fully analyzed and compared with an ANOVA approach. The article has a scientific robustness and deserve the publication but the introduction and reference sections must be improved.

Round 2
Reviewer 1 Report
Thank you for your answers to my queries.
A notable number of typing errors are to be corrected :
table 1, last line : soft liner material
line 322 the lowest
line 353 bond strength (order of the words)
line 371 percentage of failure
line 373 materials after different
line 463 failure type
line 465 dependent
line 465-470 : make 1 sentence into 2 sentences
line 378 denture base material
line 486 different
Suppress sentence lines 485-486
Reviewer 2 Report
I read the cover letter of authors and the revised paper. They extensively revised the article, especially in the parts where i identified some lack, fullfilling my requests and answering to my doubt and question.
I think that the article is now suitable for publication in Materials.
Author Response
Please see the attachment.

This manuscript is a resubmission of an earlier submission. The following is a list of the peer review reports and author responses from that submission.